# H3 Acetylation-Induced Basal Progenitor Generation and Neocortex Expansion Depends on the Transcription Factor Pax6

**DOI:** 10.3390/biology13020068

**Published:** 2024-01-23

**Authors:** Godwin Sokpor, Cemil Kerimoglu, Pauline Antonie Ulmke, Linh Pham, Hoang Duy Nguyen, Beate Brand-Saberi, Jochen F. Staiger, Andre Fischer, Huu Phuc Nguyen, Tran Tuoc

**Affiliations:** 1Department of Human Genetics, Ruhr University of Bochum, 44791 Bochum, Germany; pauline.ulmke@rub.de (P.A.U.); linh.pham@rub.de (L.P.); hoang.nguyen-j79@rub.de (H.D.N.); huu.nguyen-r7w@ruhr-uni-bochum.de (H.P.N.); 2Lincoln Medical School, University of Lincoln, Lincoln LN6 7TS, UK; gsokpor@lincoln.ac.uk; 3German Center for Neurodegenerative Diseases, 37077 Goettingen, Germany; cemil.a.kerimoglu@gmail.com (C.K.); a.fischer@eni-g.de (A.F.); 4Department of Anatomy and Molecular Embryology, Institute of Anatomy, Medical Faculty, Ruhr University Bochum, 44801 Bochum, Germany; beate.brand-saberi@rub.de; 5Institute for Neuroanatomy, University Medical Center, Georg-August-University Goettingen, 37075 Goettingen, Germany; jochen.staiger@med.uni-goettingen.de

**Keywords:** cortical development, neurogenesis, basal progenitors, gyrification, epigenetic regulation, H3 acetylation, Pax6

## Abstract

**Simple Summary:**

The mammalian cerebral cortex is believed to have gained complexity partly because of the abundance of specific cell types, collectively called basal progenitors. If these cells are deficient in the developing brain, it can lead to cortical structure anomalies, which have implications for defective brain function. Certain regulatory molecules have been found to control the production of basal progenitors during brain development. Key amongst them is the Paired Box 6 (Pax6) transcription factor and a chromatin modification mark called histone 3 lysine 9 acetylation (H3K9ac). However, our knowledge of how these regulatory factors interact to drive the generation of basal progenitors is insufficient. In this current work, we found that the enzyme involved in the acetylation of histone 3 at the 9th lysine interacts with Pax6 at the loci of genes critical for the production and amplification of the basal progenitor cell pool. As such, when both factors are decoupled or downregulated, it leads to depletion of the basal progenitor cells, resulting in a reduction in cortical development. The new mechanism identified deepens our understanding of cerebral cortex development and provides potential therapeutic cues for remedying brain abnormalities stemming from basal progenitor cell deficiency.

**Abstract:**

Enrichment of basal progenitors (BPs) in the developing neocortex is a central driver of cortical enlargement. The transcription factor Pax6 is known as an essential regulator in generation of BPs. H3 lysine 9 acetylation (H3K9ac) has emerged as a crucial epigenetic mechanism that activates the gene expression program required for BP pool amplification. In this current work, we applied immunohistochemistry, RNA sequencing, chromatin immunoprecipitation and sequencing, and the yeast two-hybrid assay to reveal that the BP-genic effect of H3 acetylation is dependent on Pax6 functionality in the developing mouse cortex. In the presence of Pax6, increased H3 acetylation caused BP pool expansion, leading to enhanced neurogenesis, which evoked expansion and quasi-convolution of the mouse neocortex. Interestingly, H3 acetylation activation exacerbates the BP depletion and corticogenesis reduction effect of Pax6 ablation in cortex-specific Pax6 mutants. Furthermore, we found that H3K9 acetyltransferase KAT2A/GCN5 interacts with Pax6 and potentiates Pax6-dependent transcriptional activity. This explains a genome-wide lack of H3K9ac, especially in the promoter regions of BP-genic genes, in the Pax6 mutant cortex. Together, these findings reveal a mechanistic coupling of H3 acetylation and Pax6 in orchestrating BP production and cortical expansion through the promotion of a BP gene expression program during cortical development.

## 1. Introduction

The brain enlarges during development and evolution. The gain in size of the neocortex during cortical growth is largely attributable to an increase in production of neurons. Excitatory neurons, which make significant contributions to neocortical parenchymal expansion, are primarily derived from apical progenitors (Aps), including apical radial glial cells (aRGCs), resident in the cortical ventricular zone (VZ). To meet the increasing neural cell number demand during cortical development, a subpopulation of Aps give rise to basal progenitors (BPs), which primarily reside in the adventricular aspect of the cortical wall, called the subventricular zone (SVZ). Since BPs amplify the neurogenic progenitor pool, the SVZ is considered as a critical neurogenic niche, which drives the increase in cortical neurogenesis [1,2,3,4,5,6]. The two major BP cell types in the developing cortex are the basal intermediate progenitor cells (bIPCs) and basal radial glial cells (bRGCs) [5,7,8,9]. For the most part, BPs can be functionally classified as proliferative or neurogenic. Neurogenic bIPCs predominate (>80%) the BP pool in the developing lissencephalic cortex of rodents, whereas proliferative bRGCs are less commonly found therein [10,11]. On the other hand, proliferative bRGCs make up a significant bulk of BPs in the embryonic gyrencephalic cortex of primates and are responsible for expansion of the cortex of such higher mammals [7,8,12]. 

Many transcription regulators have been identified to modulate the genesis and maintenance of cortical progenitors [13]. The presence and differential expression of certain transcription factors are known to confer the aforementioned interspecies variation in the abundance of cortical BPs [13]. One such example is the paired box 6 (Pax6) transcription factor, which is prominently expressed by aRGCs [14]. Indeed, functional Pax6 has been cited for its importance in cortical neurogenesis, especially in specification of BP subtypes in both gain- and loss-of-function studies [15,16,17,18,19,20,21,22].

In the absence of Pax6, intermediate progenitor cells’ (IPCs) specification is severely defective, resulting in a depleted pool of IPCs in the Pax6 mutant cortex [15,16,17,18,22]. In addition, the loss of Pax6 in a developing mouse cortex caused delamination of aRGCs to generate bRGC-like cells in a non-cell autonomous manner [15,16,17,18,19,20,21,22]. 

Proliferative BPs also have Pax6 enrichment, which seems to be important for BP subtype specification in a dose-dependent manner [16,23]. In the developing mouse cortex, when Pax6 expression was protracted in the BPs derived from Tis21-expressing aRGCs, it promoted their cell cycle re-entry, with an attendant increase in the proliferation of such BP derivatives. Notably, the sustained expression of Pax6 induced generation of primate-like bRGCs and led to increased production of upper-layer cortical neurons [23]. Similarly, it was reported that cortical neural stem cell (NSC)-specific overexpression of Pax6 (via the D6 promoter) encourages BP genesis, including T-box brain protein 2 (TBR2)-expressing BPs, which drive cortical neurogenesis [16]. Interestingly, Pax6 regulates neuronal differentiation, mostly affecting upper-layer neurons [22], via the positive regulation of pro-neural genes, including Ngn2 and Tbr2 [16,20,24,25].

Whereas it is clear that certain direct transcriptional targets of Pax6 control BP genesis [14,16], evidence is now emerging on a higher hierarchical cooperation between Pax6 and chromatin remodeling/epigenetic factors in the regulation of cortical progenitor cell generation and expansion [21]. For instance, a Pax6-BAF155 (BAF (Brg1/Brm-associated factor), an epigenetic chromatin remodeler) axis has been found critical for driving the transcriptional program that promotes the generation of BPs (bRGCs), although at the expense of neuronal differentiation. This is partly explained by the overt changes in the transcriptional and epigenetic landscape following inactivation of BAF complexes, leading to increased H3K27me3-mediated silencing of neuronal differentiation-promoting genes and downregulation of factors that maintain RG identity [21,26,27]. This observation provokes the question of whether and how Pax6 recruits epigenetic factors at Pax6 target loci to drive BP genesis. 

Our recent work reported that histone 3 (H3) acetylation is an important epigenetic regulator of cortical expansion by modulating the expression of BP-genic genes in the developing cortex [28]. In this current work, we sought to determine the contribution of H3 acetylation to the Pax6 regulatory axis, which drives BP generation and cortical expansion. We found H3K9 acetylation as a key epigenetic factor that potentiates Pax6 function in BP genesis and cortical expansion. By experimentally blocking histone 3 deacetylation with selective histone deacetylase (HDAC) inhibitors to enhance H3K9ac in the presence of functional Pax6, we were able to evoke a gyrencephalic-like cortical phenotype in the mouse neocortex hallmarked by BP pool expansion and cortical folding. However, a lack of Pax6 abolishes the BP-genic effect of H3 acetylation upregulation, leading to a loss of upper-layer neurons and a reduction in cortical size. Our data suggest that the control of BP genesis by H3 acetylation involves activation of Pax6 transcriptional activity. The findings highlight the dependence of H3K9ac on Pax6 functionality as a mechanism for eliciting developmental and likely the evolutionary expansion of the mammalian cortex. 

## 2. Materials and Methods

### 2.1. Animal Care and Procedures 

All mouse lines (floxed *Baf155*, floxed *Pax6*, and *Emx1-Cre* [29,30,31]) were generated and maintained in a C57BL6/J background. In utero electroporation was carried out using a previously described routinely established protocol [27,32,33]. Pharmacological treatment of mice with the HDAC inhibitor (HDACi) was performed, as previously described [28]. Animal handling was in accordance with the German Animal Protection Law and with the permissions (14/1636, 16/2330, and 18/3038) of the Bezirksregierung Braunschweig, according to the Institutional Animal Care and Use Committee guidelines. 

### 2.2. Plasmids

Plasmids used in this study are as follows: *pCON-P3-Luc* (2xP6CON plus 3xP3 sequences in pGL3 basic, Promega, as described in [32]), *pLuc-Cux1* [34], *pLuc-Ngn2* [25] and as a gift from Dr. Francois Guillemot, NIMR London, *pLuc-Tbr2* [35], as a gift from Dr. Miyata, Nagoya University, *CMV/Pax6*, and *CMV/Kat2a*.

### 2.3. Antibodies

The following are the polyclonal (pAb) and monoclonal (mAb) primary antibodies used in this study: Pax6 rabbit pAb (1:200; Covance, Schwerte, Germany), H3K9Ac rabbit pAb (Abcam, Cambridge, UK), Sox2 goat (1:100; Santa Cruz, Dallas, TX, USA), Sox2 mouse mAb (1:100; R&D Systems, Minneapolis, MN, USA), Pax6 mouse mAb (1:100; Developmental Studies Hybridoma Bank, MA, USA), Tbr2 rabbit pAb (1:200; Abcam), H3ac rabbit pAb (Upstate, MA, USA), TNC rabbit pAb (Abcam), AP2γ mouse mAb (1:100; Abcam), PTPRZ1 rabbit pAb (Sigma, St. Louis, MO, USA), BAF155 mouse mAb (1:100; Santa Cruz), BAF155 rabbit pAb (1:20; Santa Cruz), Tbr1 rabbit pAb (1:300; Chemicon, St. Louis, MO, USA), HuCD mouse mAb (1:20; Invitrogen, Schwerte, Germany), Ctip2 rat pAb (1:200; Abcam), Cux1 rabbit pAb (1:100; Santa Cruz), Satb2 mouse mAb (1:200; Abcam), NeuN mouse mAb (Chemicon), Nestin mouse mAb (BD, Heidelberg, Germany), and Kat2a rabbit pAb (Abcam).

Secondary antibodies used were horseradish peroxidase (HRP)-conjugated goat anti-rabbit IgG (1:10,000; Covance), HRP-conjugated goat anti-mouse IgG (1:5000; Covance), HRP-conjugated goat anti-rat IgG (1:10,000; Covance), and Alexa 488-, Alexa 568-, Alexa 594-, and Alexa 647-conjugated IgG (various species, 1:400; Molecular Probes, Schwerte, Germany). Antibodies were purchased from the indicated manufacturers in Germany.

### 2.4. Fluorescence-Activated Cell Sorting (FACS)

Tbr2-expressing BPs were purified from the developing mouse cortex using a FACS technique previously used for cell sorting [28,36,37]. The adopted FACS method was used to effectively isolate Tbr2-expressing BPs, mainly bIPCs, in the E15.5 WT and *Pax6cKO* mouse cortex. 

### 2.5. Chromatin Immunoprecipitation (ChIP)

Embryonic mouse cortices were homogenized in sucrose solution (0.32 M sucrose, 5 mM Mg(Ac)_2_, 5 mM CaCl_2_, 50 mM HEPES pH 8, 1 mM DTT, 0.1% Triton X-100, and 0.1 mM EDTA), fixed in 37% formaldehyde, and then treated with 1.25 M glycine. The samples were then centrifuged, and the pellet (i.e., nuclei) was washed with Nelson buffer. Chromatin fragments were prepared, immunoprecipitated, and analyzed, as previously described [28,38].

We tested the following antibodies for KAT2a ChIP (NBP1-00845, Novus Biologicals; ab1831, Abcam; 07-1545, Millipore, MA, USA; 3305, Cell Signaling; sc-20698 (H-75), Santa Cruz; 607201 Biolegend) [39], Pax6 ChIP (PRB-278P, Covance; AB2237, Chemicon), and BAF155 ChIP (sc-10756, Santa Cruz; ab126180, Abcam), but none of the antibodies showed any difference from the IgG control and were, therefore, considered not suitable for ChIP experiments.

### 2.6. ChIP-Seq

The NEBNext Library Prep Kit for Illumina (NEB) was used to prepare libraries. Isolated DNA (100 ng as input) was pooled for each group. A QuBit fluorometer and Agilent 2100 Bioanalyzer were used to measure the quantity and quality of the libraries, respectively. 

Base calling and conversion to fastq format were performed using Illumina pipeline scripts. Quality control was then conducted on raw data for each library [FastQC, www.bioinformatics.babraham.ac.uk/projects/fastqc]. Typical analytical information and control measurements were obtained. STAR aligner v2.3.0 [40] was used to map reads to a mouse reference genome (mm10). PCR duplicates were removed from each BAM file using the rmdup function of samtools [41]. The merge function of samtools allowed a single merger of BAM files belonging to same-group replicates. All downstream analyses were performed on BAM files with only unique reads.

H3K9ac profile plots were created with NGSPlot [42] using merged BAM files from immunoprecipitated samples and inputs. The paired Student’s *t*-test was performed on the data used for generating the plots. By means of the Integrated Genome Browser, H3K9ac enrichment was visualized at different gene loci [43] using wiggle files that were created from the merged BAM files with the script from the MEDIPS package of Bioconductor [44]. 

Peaks were called on individuals with MACS2 with a q-value < 0.1 [45]. Assessment of differential binding was performed using the DiffBind package of Bioconductor [46]. HOMER was used for peak annotation [47]. 

### 2.7. RNA-Sequencing

RNA was prepared (RNeasy kit; Qiagen) from the E15.5 cortex. The TruSeq RNA Sample Preparation v2 Kit was used to prepare cDNA libraries. Quantity and quality of DNA were measured using a Nanodrop spectrophotometer and an Agilent 2100 Bioanalyzer, respectively. 

Quality control, fastq conversion, base calling, and read alignments were performed, as for ChIP-Seq. FeaturesCount (http://bioinf.wehi.edu.au/featureCounts/) was used for aligning and counting reads. Differences in gene expression were assessed using DESeq2 from Bioconductor [48]. Gene enrichment analyses were performed using ToppGene [49]. All equipment and software were sourced or purchased in Germany.

### 2.8. Immunohistochemistry (IHC) Experiment

IHC experiments were performed using routinely established methods, as previously described [22]. 

### 2.9. qRT-PCR and WB Analyses

qRT-PCR and WB analyses were performed as previously reported [32] using primers listed in Appendix A.

### 2.10. Luciferase Assay

Neuro2A cells were cultured, transfected with plasmids of interest, and lysates were analyzed for luciferase activity, as previously described [21,34].

### 2.11. Assay for Pax6 Interaction Protein (Yeast Two-Hybrid Screening, WB, coIP)

To identify Pax6 cofactors, we employed a yeast two-hybrid assay to screen a mouse embryonic (E15.5) cortical cDNA library, using *Pax6∆PD* (PD domain-deleted Pax6) as a bait (refer to [32] for full method description). Out of a library containing 2.5 × 10^6^ transformants, 4 sequences obtained from a total of 116 independent clones were found to code for KAT2A/GCN5, an enzyme known to acetylate H3K9 at promoters and enhancers [50,51,52]. To confirm that Kat2a binds to Pax6, we performed coimmunoprecipitation (coIP) assays using the Pax6 antibody or IgG as a negative control and lysates of the E15.5 cortex. Western blot analysis using the KAT2A antibody was performed to observe the physical interaction between Pax6 and KAT2A. The lack of binding between IgG and KAT2A was taken as a control.

### 2.12. Cell Counts and Quantitative Analysis of Immunohistochemical Signal Intensity

Signal intensities of H3K9ac and H3ac were quantified in immunohistochemical micrographs of mouse cortex sections. The colored stain signals were converted to grayscale to eliminate the background. The pixel values of the fluorescent signal intensity were measured with ImageJ [34,53]. Similarly, the relative amount of protein from developed films in the WB experiment was quantified densitometrically using ImageJ software. Details of the methods can be found in [32,34,38,53].

Anatomically similar forebrain sections/regions from control and mutants or HDACi-treated embryos were selected for quantification and comparison. In the majority of cell counting, six matched sections were averaged from biological triplicates. 

### 2.13. Imaging and Statistical Analysis

All micrographs were obtained using standard (Leica DM 6000) and confocal fluorescence (Leica TCS SP5) microscopes, or an Axio Imager M2 (Zeiss) with a Neurolucida system (MBF Bioscience). Images were further analyzed with Adobe Photoshop. Statistical analyses were performed using Student’s t-test. All bar graphs are plotted as means ± SEM. All statistical tests are two-tailed, and *p*-values are considered significant for α = 0.05. Details of the statistical analyses for histological experiments are presented in Appendix A.

## 3. Results

### 3.1. Pax6 Is Indispensable for H3 Acetylation-Induced bIPC Production

Work from our group and others indicates that the transcription factor (TF) Pax6 is an intrinsic determinant of aRGCs, promoting bIPC specification involved predominantly in the generation of neurons with upper-layer identities [15,16,17,18,22,23]. Across species, Pax6 expression in neural progenitors is downregulated during differentiation of aRGCs to bIPCs [23,54,55,56]. Such Pax6 downregulation is prominent in non-primate BPs compared with BPs found in the primate(-like) neocortex [8,10,12,54,57,58]. Interestingly, Wong et al. found that when Pax6 expression is sustained in cortical progenitors it promotes the generation of primate-like BPs in the mouse cortex [23]. These previous observations highlight a critical role for Pax6 in BP biogenesis during cortical ontology and evolution.

We previously showed that the cortex-specific loss of BAF155, as seen in BAF155cKO embryos, promoted delamination of APs, increasing the population of PAX6+, TBR2+ BPs in the intermediate zone (IZ) [21]. As PAX6+ BPs are relatively rare in the WT mouse cortex, we used the BAF155cKO mutant as a mouse model to investigate the effect of HDAC inhibition on the proliferation of BPs. Furthermore, we found that increased H3 acetylation augments genesis of BPs, including both bIPCs and bRGCs [28]. The enhanced genesis of BPs in response to elevated H3 acetylation was observed in both the WT and *BAF155cKO* cortex treated with H3 acetylation-increasing agents [28]. Given the prominent role of Pax6 in the generation of neurogenic progenitors, we followed up by further asking whether enhanced H3 acetylation also leads to an increase in BP production in the absence of Pax6 expression (i.e., in the Pax6 mutant cortex). 

To address this question, we administered Trichostatin A (TSA)—a selective class I/II HDACi—to mutant embryos lacking both BAF155 and Pax6 (*BAF155/Pax6dcKO*). Control (WT) and *BAF155cKO* embryos were given similar treatment. The pharmacological treatment with TSA commenced 12.5 days post coitum (d.p.c.) and embryos were examined from E16.5 to E18.5 (Figure 1A) [28]. We observed that inhibition of H3 deacetylase activity via TSA treatment (i.e., elevation of H3 acetylation) caused an increase in the basally located Tbr2+ IPCs and those in transition between aRGCs and bIPCs (AP2γ+ cells) in the E16.5 neocortex of WT and *BAF155cKO*-expressing Pax6 (Figure 1B,C). Consistent with a previous observation in other Pax6 mutants, such as Sey/Sey and cortex-specific *Pax6cKO* mutants [15,16,17,18,22], the absence of Pax6 led to reduced numbers of neurogenic IPCs in the vehicle (Veh)-treated *BAF155/Pax6dcKO* mutant cortex when compared with the WT cortex under similar experimental conditions (Figure 1B,C). Interestingly, TSA treatment rendered the *BAF155/Pax6dcKO* mutant cortex, especially in the SVZ, even more deprived of Tbr2-expressing bIPCs when compared with the untreated mutant cortex (Figure 1B,C). Along a similar line of observation, HDAC inhibition via TSA treatment caused an increase in both apical and basal AP2γ-expressing cells in the WT and *BAF155cKO* developing mouse brain. However, TSA-mediated HDAC inhibition resulted in a marked reduction in the quantity of such AP2γ-expressing cells in the *BAF155/Pax6dcKO* cortex (Figure 1B,C). Consistent with the effects evoked by TSA treatment, we observed that treatment of the developing mouse cortex with other HDACi agents, namely valproic acid (VPA) (Figure 1D) and suberoylanilide hydroxamic acid (SAHA) (Figure 1E), recapitulated the enhanced BP genesis phenotype.

This set of results indicates that Pax6 may play a critical role in H3 acetylation-mediated generation of neurogenic IPCs. 

### 3.2. Pax6 Expression Is Required for H3 Acetylation-Induced Genesis of bRGCs

Normally, bRGCs are seldom found in the mouse cortex [10]. Previously, we compared the gene expression profile of TSA- and vehicle-treated *BAF155cKO* cortices via RNA-seq [28]. Interestingly, TSA treatment led to an increased expression of genes enriched in bRGCs (e.g., TNC, PTPRZ1, PAQR8, Gigyf2, PDLIM3, and ZC3HAV1) [58,59]. Following TSA treatment, qPCR analysis confirmed the increased RNA levels of the aforementioned bRGC-enriched genes in the WT mouse cortex. We observed even higher levels of expression of these bRGC genes in the TSA-treated *BAF155cKO* cortex (Figure 2A). Nonetheless, eliminating Pax6 in the *BAF155cKO* mutant cortex (i.e., yielding *BAF155/Pax6dcKO*) rendered the double-mutant cortex unresponsive to H3 acetylation-mediated upregulation or induction of bRGC gene expression, and levels remained basal, as in the controls (Figure 2A). 

We then employed immunohistochemical staining using antibodies against some selected bRGC-enriched genes, TNC and PTPRZ1 (Figure 2B–D), to link bRGC cellular presence to the observed increase in bRGC-enriched gene expression. Indeed, we observed an increased immunofluorescent signal of TNC and PTPRZ1 both in VZ and IZ in the WT cortex and much more intense in the *BAF155cKO* cortex treated with TSA (Figure 2B–D). However, in the absence of Pax6, as in the treated *BAF155/Pax6dcKO* cortex, TNC or PTPRZ1 immunostaining is overtly reduced when compared to the treated WT or *BAF155cKO* cortex (Figure 2B–D). To compare the population of aRGCs in VZ and bRGCs between cortexes from the above-treated animals, we performed double IHC analysis for either TNC (Figure 2C) or PTPRZ1 (Figure 2D) with antibodies against the NSC nuclei marker Sox2. Of note, some of the Sox2-expressing basal NSCs co-express the bRGC markers TNC (Figure 2C) and PTPRZ1 (Figure 2D). Such Sox2/TNC or Sox2/PTPRZ1 colocalizations were found to be quantitatively more in the *BAF155cKO* cortex, to a lesser extent in the WT treated with TSA (Figure 2E), and very rare in the Veh-treated WT. Interestingly, almost no Sox2+/TNC+ or Sox2+/PTPRZ1+ bRGC was observed in the TSA-treated *BAF155/Pax6dcKO* (Figure 2C–E).

Together, these results show that bRGC production is a major consequence of H3 acetylation activation, which leads to expansion of the bRGC pool in the developing mouse cortex, known to normally have a scanty amount of bRGCs [10]. However, such H3 acetylation-induced bRGC generation depends on normal Pax6 functionality.

### 3.3. H3 Acetylation Drives Cortical Expansion and Folding in a Pax6-Dependent Manner

The phenomenon of cortical expansion and folding, which characterizes the brain of large primates, is primarily attributed to the abundance of BPs known to amplify the neuronal output of primary NSCs during corticogenesis [10,60,61,62]. Since augmented H3 acetylation leads to an increase in the generation of bIPCs and bRGCs in the neocortex of WT and *BAF155cKO* treated with TSA (Figure 1 and Figure 2), we sought to determine the implication of BP pool expansion in the HDACi-treated WT and *BAF155cKO* cortex in terms of cortical growth and morphology [28]. The BP pool-depleted, HDACi-treated *BAF155/Pax6dcKO* cortex was also examined using similar phenotype indices to assess the contribution of Pax6 in cortical expansion. 

Based on immunohistochemical staining of various populations of cortical neurons to outline cortical layers, it was observed that TSA treatment caused folding of the *BAF155cKO* cortex, but to a mild and infrequent extent in the WT cortex (Figure 3A) [28]. While two-thirds of all the examined TSA-treated *BAF155cKO* brains (*n* = 9) presented an intensely folded cortex and the remaining showed mild cortical folding, less than half (4 out of 9) of the examined TSA-treated WT cortices were mildly folded, and only one had an intensely folded cortex (Figure 3B). However, as observed in the vehicle-treated WT brain (control), TSA treatment did not cause cortical folding in the absence of Pax6 in all the examined *BAF155/Pax6dcKO* brains (Figure 3B).

It is worth noting that the presence and orientation of RGC fibers contribute to cortical folding by providing the gyrification mechanics, including acting as scaffolds for differential placement of migrating neurons [6,63]. The induction of bRGCs following H3 acetylation upregulation in Pax6-expressing aRGCs (Figure 2) implies an increase in RGC fibers, which may partly underscore the cortical folding propensity observed in the mutants (Figure 3A,B).

By quantifying the area of the Satb2 immunosignal, it was identified that the *BAF155cKO* and WT cortexes treated with TSA had a larger size when compared with controls. However, TSA treatment resulted in more cortical enlargement in the *BAF155cKO* than in WT (Figure 3C). On the other hand, the Veh-treated *BAF155/Pax6dcKO* cortex displayed a reduced cortical size when compared with all the other experimental groups. An interesting observation was the increase in the number of cortical (pallial) cells expressing NeuN (marker for both pyramidal neurons and interneurons) in the TSA-treated *BAF155/Pax6dcKO* cortex, as compared to the vehicle-treated *BAF155/Pax6dcKO* cortex (Figure 3D). This effect can be linked to the slight increase in the Satb2-marked cortical area in the *BAF155/Pax6dcKO* following TSA treatment, as compared to without TSA treatment (Figure 3C). The increase in NeuN+ cells in the TSA-treated *BAF155/Pax6dcKO* is not accounted for by neurons expressing Ctip2 or Satb2, which are mainly derived from BPs, since such neurons are reduced in the TSA-treated *BAF155/Pax6dcKO*, as compared with the vehicle-treated *BAF155/Pax6dcKO* cortex (Figure 3D). 

Together, these results indicate that the increase in cortical neurogenesis is an effect of HDACi treatment (H3 acetylation upregulation) and is preceded by an expansion of the BP pool. However, this effect is contingent on Pax6 expression, since in the absence of Pax6, the BAF155 mutant cortex (*BAF155cKO*) loses its ability to produce more neurons, affording the observed cortical size expansion and gyrification (Figure 3A,B).

### 3.4. Global Impairment of the Euchromatin Mark H3K9ac and Gene Expression Programs in Pax6-Deficient Developing Cortex

Lack of BP generation in the absence of Pax6 in the *Pax6 mutant* cortex in both vehicle and HDACi treatment (increased H3 acetylation) prompted us to examine the link between Pax6 and H3K9 acetylation in regulation of BP gene expression. The cortex-specific *Pax6cKO* mutant mouse brain [22] was profiled for changes in H3K9 acetylation. Chromatin immunoprecipitation against H3K9ac, followed by deep sequencing (ChIP-seq), revealed a drastic reduction in H3K9ac in the embryonic cortex of *Pax6cKO* mutants (Figure 4A, Appendix A). While 2531 genes showed a significant decrease (*p*-value < 0.01 and |fold change| > 1.2) in H3K9ac at their promoters, not a single gene promoter region had elevated H3K9ac in the *Pax6cKO* cortex (Figure 4B). 

In RNA-seq experiments (Appendix A), we found that neurogenic genes that are specifically expressed in bIPCs (e.g., Tbr2), or in cells transitioning between aRGs and bIPs, and in neurons (e.g., *Ngn1*, *Ngn2*, *Cux1*, *Cux2*, and *NeuroD1*) [35,64,65,66,67,68,69,70,71], were downregulated in the *Pax6cKO* cortices [21]. Importantly, compared to the control, a lower level of H3K9ac at the loci of these genes, especially IPC genes, was also observed in the cortex of the *Pax6* mutants (Figure 4C,D). As exemplified in Figure 4E, the distribution of H3K9ac marks associated with gene promoters, such as Ngn1, Ngn2, Tbr2, Cux1, and Cux2, was demonstrably reduced in the E15.5 cortex of *Pax6* mutants (Figure 4E, Appendix A). 

Together, these findings suggest that deletion of Pax6 in the developing cortex leads to a lack of H3K9ac-linked activation of genes important for BP specification/generation. 

### 3.5. Pax6 Interacts with Kat2a to Cause H3K9ac-Linked Activation of BP Genes

Next, we sought to evaluate whether Pax6 controls the expression and promoter level H3K9ac of its target genes (e.g., Ngn2, Tbr2, and Cux1) in bIPCs. To this end, the FACS method was used to isolate Tbr2-expressing BPs in the E15.5 WT and *Pax6cKO* cortexes (Figure 5A–C).

We used H3K9ac ChIP analysis followed by qPCR to measure the level of H3K9ac marks associated with three selected neurogenic and Pax6 target genes (*Ngn2*, *Tbr2*, and *Cux1*) expressed in the sorted bIPCs. Compared to the WT (control), the levels of H3K9ac at the promoter regions of *Ngn2*, *Tbr2*, and *Cux1* genes were reduced in the sorted Tbr2+ IPCs lacking in Pax6 (i.e., in the *Pax6cKO* cortex) (Figure 5D). In addition, the relative levels of *Ngn2*, *Tbr2*, and *Cux1* transcripts in the *Pax6* mutant Tbr2+ IPCs were reduced by more than half, as compared with levels in the control Tbr2-expressing IPCs (Figure 5E). 

Previously, we searched for cofactors that may modulate the neurogenic function of Pax6 in cortical neurogenesis by screening for Pax6 interacting partners [32]. By means of a protein interaction assay, we found KAT2A/GCN5, an enzyme known to acetylate H3K9 at promoters and enhancers [50,51], to interact with Pax6 (Figure 5F).

Kat2a is considered an important regulator of gene expression because its ablation results in H3K9 depletion and downregulation of the transcriptional program [72]. Based on this line of reasoning, we tested how the presence or absence of the HDAC inhibitor TSA and H3K9 acetyltransferase Kat2a influence the Pax6-dependent transcriptional activity and promoter activity of Pax6 target genes. To carry out this investigation, we assayed luciferase activity in cultured neurons (Neuro2A) co-transfected with luciferase (Luc) plasmids of *Pax6* target gene promoters with plasmids containing *Kat2a* and *Pax6* under various experimental conditions of TSA treatment in a combinatorial manner, as depicted in Figure 5G. The luciferase measurements (Figure 5G) revealed that Pax6, Kat2a, and H3 acetylation regulate the transcriptional activity of IPC genes whose expressions are controlled by Pax6. In detail, we observed increased luciferase activity in neurons transfected with Kat2a and/or Pax6. Whereas a lack of Pax6 alone is sufficient to cause a loss of luciferase activity in the cultured neurons, an additional treatment with TSA augmented such luciferase activity (Figure 5G). Together with the outcome of co-transfections with luciferase plasmids containing cloned DNA fragments of Pax6 binding site(s) of the indicated Pax6 target genes (Figure 5G), we were able to determine that Pax6 cooperates with Kat2a and, hence, H3 acetylation to transcriptionally activate neurogenic genes.

## 4. Discussion

The neurodevelopmental complexification of the brain is hallmarked by cortical expansion and gyrification, which underscores the acquisition of high cognitive functions. Various mechanisms or factors have been identified as key players in cortical expansion and convolution. These include genetic, cellular, migration, and biomechanical factors [73,74,75,76,77,78]. 

Several studies show a strong association between BP enrichment and increases in the propensity of the cortex to convolute [79,80,81,82,83,84,85,86]. As a regulator of neural progenitor cell multipotency, proliferation, and differentiation in the central nervous system, the transcription factor Pax6 is recognized as a key player in the expansion of the neurogenic progenitor pool and, hence, neurogenesis. The differential expression of Pax6 ensures a balance between the proliferation (self-renewal) and differentiation of neural stem cells to determine cortical growth [16]. To stimulate BPs’ generation and pool expansion in the developing cortex, Pax6 expression is sustained in BP-genic progenitor cells and their progenies (BPs). Such protracted expression of Pax6 by neural progenitor cells is known to promote increased proliferation of BPs due to cell cycle re-entry [23]. Our recent studies [28], and the current one, add a critical layer of mechanisms for how BP pool expansion occurs to afford cortical growth. We have identified that Pax6 interacts with the H3 acetyltransferase Kat2a to drive an increase in H3 acetylation at the promoter of BP genes, resulting in the potentiation of BP genesis and a concomitant increase in cortical size (Figure 6).

### 4.1. Cortical BP Pool Expansion Depends on H3 Acetylation in the Presence of Pax6

A major difference between the cortex of highly evolved primates and non-primates (with a smooth cortex) is that the former is enriched with BPs compared to the latter. Could the basis for this difference be partly due to the central finding of this study, wherein we observed that H3 acetylation enhances the proliferation and/or generation of BPs in a Pax6-dependent manner? We posit that H3 acetylation is a major means by which BP enrichment is achieved in the developing cortex because its upregulation triggers an overt increase in the Tbr2-expressing bIPCs and Pax6-expressing bRGC-like cells. This conclusion is supported by the observation that Pax6 is highly expressed by BPs in the primate cortex (reviewed in [14]), which parallels the identified upregulation of H3K9ac in the developing human cortex (this study; [28]). The striking upregulation of bRGC genes in the HDACAi-treated mouse cortex, which are preferentially/highly expressed in the primate cortex, shows that enforced H3 acetylation conditioned BP-genic aRGCs to produce more bRGC-like cells and demonstrably diversified the mouse BP pool to mimic that of gyrencephalics [87]. 

Although deletion of BAF155 alone was reported to promote the generation of bRGCs in the mouse cortex [21,27], there is a lack of BPs in the *dcKO_BAF155_Pax6*, even in the presence of elevated H3 acetylation, suggesting that Pax6 functionality is a determinant of H3 acetylation-dependent BP genesis. It is conceivable that the treatment of the *BAF155 cKO* cortex with HDACi led to upregulation of H3 acetylation in the induced bRGCs, leading to an increase in their proliferation capacity and promoting generation of bIPCs that express Pax6. Of note, in the absence of enforced H3 acetylation, bIPCs are depleted in the *BAF155 cKO* cortex, albeit there is an induction of bRGCs in such a mutant cortex [21,27]. The explanation is that the induced bRGC-like cells in the *BAF155 cKO* cortex are generated by unregulated delamination of Pax6-expressing aRGCs due to a loss of apical adhesive proteins [21,27]. However, because of the downregulation of native H3 acetylation in the cortex of BAF155 mutants, the induced bRGCs are less proliferative or lack the ability to generate bIPCs. It is only after exogenous stimulation of H3 upregulation in the BAF155 mutant cortex that the induced bRG-like cells gain increased capacity to proliferate and produce more bIPCs. Nonetheless, the augmented BP pool phenotype is abolished when Pax6 is inactivated; thus, we put forward that Pax6 occupies a pivotal position in the regulation of BP genesis in response to H3 acetylation upregulation.

### 4.2. Pax6 Interacts with Kat2a to Activate H3K9ac-Dependent Gene Expression Required for BP Genesis and Cortical Expansion

Having established evidence for the contribution of BP abundance, especially bRGCs, to the folding of the cortex [75], there is growing interest in describing subtle molecular mechanisms that drive expansion of the BP pool during cortical development. Given that we observed an increase in the BP pool following treatment of the mouse cortex with HDACi, for which the phenotype becomes marked in the presence of Pax6, we focused on defining a mechanism linking H3 acetylation to Pax6 functionality in cortical growth. 

Identifying Kat2a, in this study, as an interaction partner of Pax6 consolidated the assertion that H3 acetylation requires Pax6 in targeting regulatory elements of BP-genic or neurogenic genes for H3K9ac upregulation. Indeed, the luciferase experiment revealed the transcriptional activity of such genes to increase under the conditions of Pax6 and Kat2a or HDACi treatment. The singular effect of Pax6 overexpression leading to an increase in activity of the said genes was expected and consistent with a previous report of sustained Pax6 expression promoting the generation of BPs during cortical development [23]. Interestingly, the mutant cortex with Nestin_cre-mediated conditional knockout of *Kat2a* phenocopies that of Pax6 deletion in the cortex and presents a loss of apical cell adhesion proteins with concomitant delamination of aRGCs, leading to ectopic generation of BPs, including bRG-like cells [19,21,87,88].

Put together, we described a previously unknown mechanism involving Pax6, recruiting or interacting with the H3 acetyltransferase KAT2A to upregulate the installation of H3Kac marks to activate the transcription of downstream factors required for the amplification of BP genesis and possible BP pool diversification (Figure 6). The uncovered mechanism elaborates the previously described mechanism for BP generation involving the interaction between BAF155 and Pax6 [34,53]. Thus, Pax6 binding at regulatory elements of its gene targets may elicit a BP-genic transcriptional program via coordinating the configuration of the chromatin landscape, featuring chromatin remodelers (e.g., BAF complex) and/or H3 acetylation modules, which support amplification of the BP pool and promotion of neurogenesis, as partly identified for some pro-neural factors [89,90]. 

As expected, the abundance of BP translated into an increase in neurogenesis, and the increase in neuronal output manifested as marked cortical growth/expansion, which triggered the development of cortical folds normally uncharacteristic of the mouse cortex. Our attention was drawn to the moderate increase in the total number of neurons (NeuN+ cells) in the *dcKO_BAF155_Pax6* cortex treated with HDACi compared with the HDACi-untreated *dcKO_BAF155_Pax6* cortex. Since such an increase in the neuronal population is not accounted for by the BP-derived neuronal subtype (i.e., later-born SATB2-expressing), we can explain the increase in neuronal cells in the HDACi-treated *dcKO_BAF155_Pax6* cortex as emanating from the presence of other neuronal cell types, say, those derived from the ventral telencephalon (i.e., interneurons derived from the ganglionic eminences [30,91,92,93,94]), in the mutant cortex. Our further investigations in a separate study point to a possible fate change effect of Pax6 ablation, leading to neocortical neural stem cells acquiring the ability to generate interneurons upon HDACi treatment or H3 acetylation upregulation. It implies that Pax6 and H3K9ac enrichment plausibly increases the multipotency of neural stem cells in the developing neocortex. 

## 5. Conclusions

The current work has unraveled a mechanistic cooperation between H3 acetylation, via Kat2a involvement, and Pax6 function in orchestrating BP generation, leading to developmental expansion of the mammalian cortex. The upregulation of H3 acetylation and Pax6 expression in cortical progenitors found in the primate or evolved cortex is suggestive of these factors contributing to the evolutionary expansion and ramifications of the cerebral cortex. Thus, this finding fills an important gap in our knowledge of the mechanistic basis of BP pool expansion, which drives cortical growth and expansion. 

## Figures and Tables

**Figure 1 biology-13-00068-f001:**
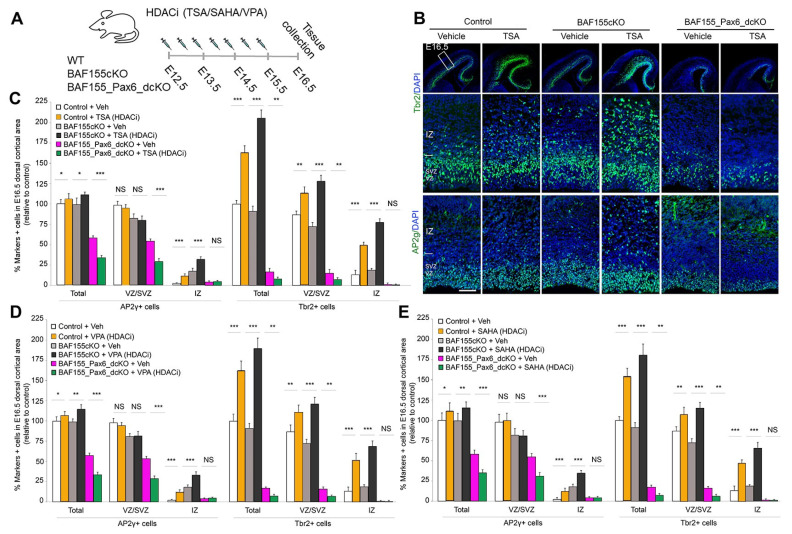
H3 acetylation-induced genesis of bIPCs in the developing cortex depends on Pax6 presence. (**A**) Experimental scheme in which control and mutant embryos were treated with the HDAC inhibitor. (**B**) IHC micrographs showing Tbr2 (bIPs marker) and AP2ɣϫ (marker for APs and BPs) staining in mouse cortical sections from wild-type, *BAF155cKO*, and *BAF155_Pax6_dcKO* embryos with or without TSA treatment. Counter staining with DAPI is shown. Modified from Figure 3B in [28]. (**C**–**E**) Bar graphs showing changes in the numbers of apical and basal progenitor cells in the indicated cortical wall regions following treatment of embryos with the HDAC inhibitor TSA (**C**), VPA (**D**), or SAHA (**E**). Values are presented as means ± SEMs (* *p* < 0.05, ** *p* < 0.01, *** *p* < 0.005, NS, not significant; *n* = 6). Abbreviations: TSA, trichostatin A; SAHA, suberoylanilide hydroxamic acid; VPA, valproic acid; VZ, ventricular zone; SVZ, subventricular zone; IZ, intermediate zone. Scale bars = 50 µm.

**Figure 2 biology-13-00068-f002:**
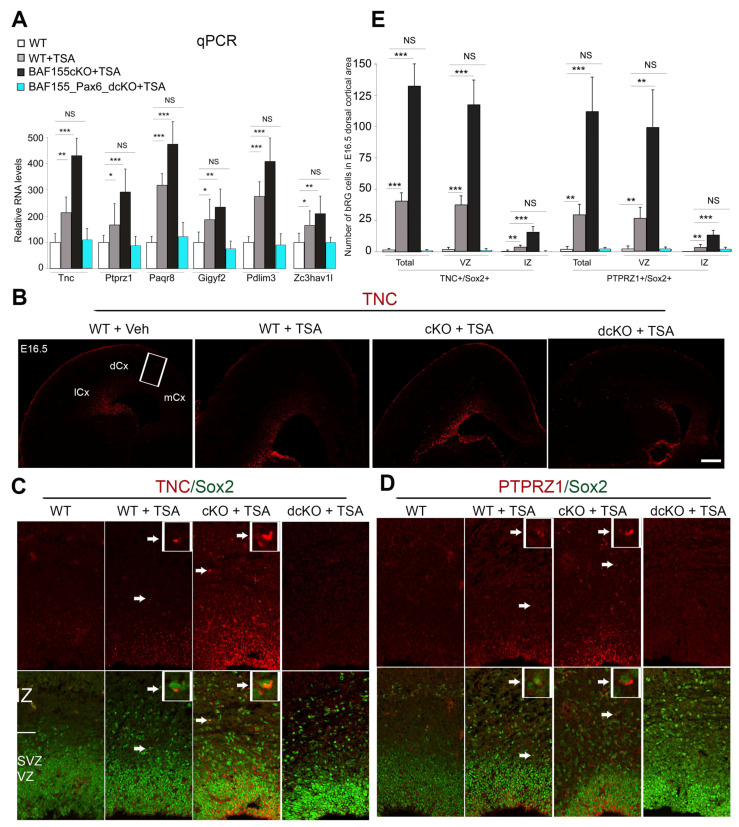
HDAC inhibition induces generation of bRGCs in a Pax6-dependent manner. (**A**) Bar graphs showing the qPCR analysis used to confirm the upregulated expression of human-enriched bRGC markers in the TSA-treated *BAF155cKO* cortex and downregulation in the *BAF155_Pax6_dcKO* cortex, as compared with the indicated controls. (**B**–**D**) Micrographs showing immunostaining of selected human-enriched bRG markers (TNC and PTPRZ1) and Sox2 in the TSA-treated *BAF155cKO* and *BAF155_Pax6_dcKO* cortices compared with the indicated controls. Arrows point to cells co-expressing TNC and Sox2 or PTPRZ1 and Sox2. White rectangle in (**B**) indicates sampled cortical area imaged at high power in (**C**,**D**). Rectangular inserts in (**C**,**D**) are zoomed in images of the arrowed immunosignals. (**E**) Bar graphs showing quantitative analysis of cells immunostained for the bRG markers PTPRZ1 and TNC and their co-expression of Sox2 in the TSA-treated cortices compared with the untreated/vehicle-treated cortex. Values are presented as means ± SEMs (* *p* < 0.05, ** *p* < 0.01, *** *p* < 0.005; NS, not significant; *n* = 6). Abbreviations: l/d/mCx, lateral/dorsal/medial cortex; VZ, ventricular zone; IZ, intermediate zone. Scale bars = 100 μm.

**Figure 3 biology-13-00068-f003:**
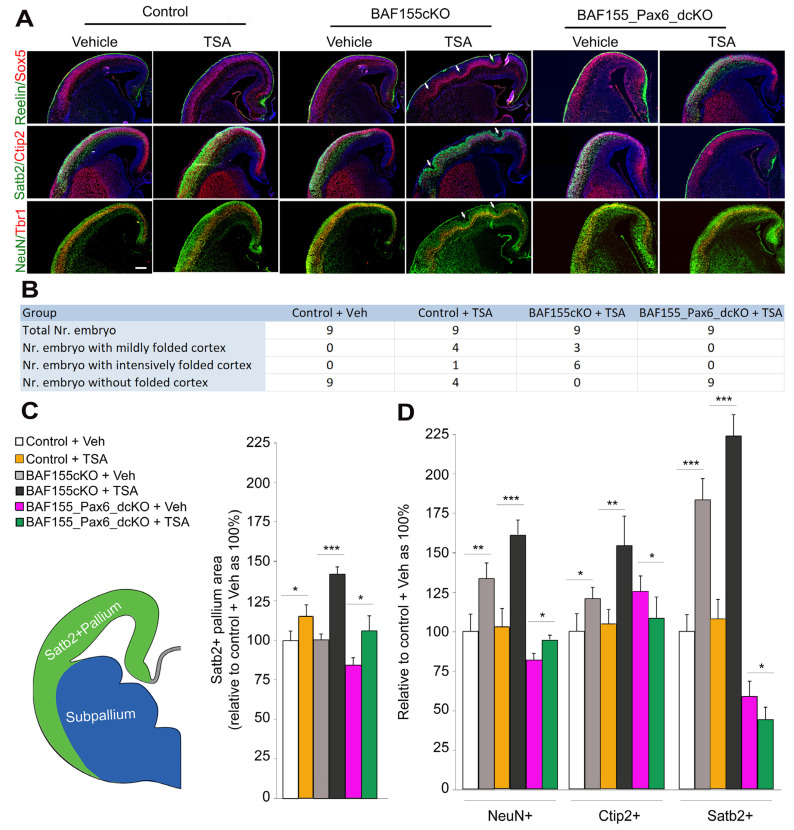
Elevated H3 acetylation promotes cortical expansion and folding with Pax6 expression involvement. (**A**) IHC micrographs showing the developing mouse cortex stained with the indicated antibodies to reveal profiles of cortical layers and cell populations as an indication of cortical gyrification following treatment of wild-type, *BAF155cKO*, and *BAF155_Pax6_dcKO* brains with TSA. Arrows point at putative sulci. Counter staining with DAPI is shown. Modified from Figure 7C in [28]. (**B**) Image showing tabulations of the number of embryos with a folded cortex and the extent of folding following treatment of the wild-type, *BAF155cKO*, and *BAF155_Pax6_dcKO* brains with TSA. (**C**) Image showing the quantification of the Satb2-positive cortical/pallial area in the wild-type, *BAF155cKO*, and *BAF155_Pax6_dcKO* cortexes treated with TSA, as compared with the indicated controls. (**D**) Bar graph showing quantification of all neurons (NeuN-expressing cells), lower-layer neurons (Ctip2-expressing cells), and upper-layer neurons (Satb2-expressing cells) in the wild-type, BAF155cKO, and *BAF155_Pax6_dcKO* cortexes treated with TSA, as compared with the indicated controls. Values are presented as means ± SEMs (* *p* < 0.05, ** *p* < 0.01, *** *p* < 0.005; *n* = 6). Abbreviations: Veh, vehicle; TSA, trichostatin A. Scale bars = 100 μm.

**Figure 4 biology-13-00068-f004:**
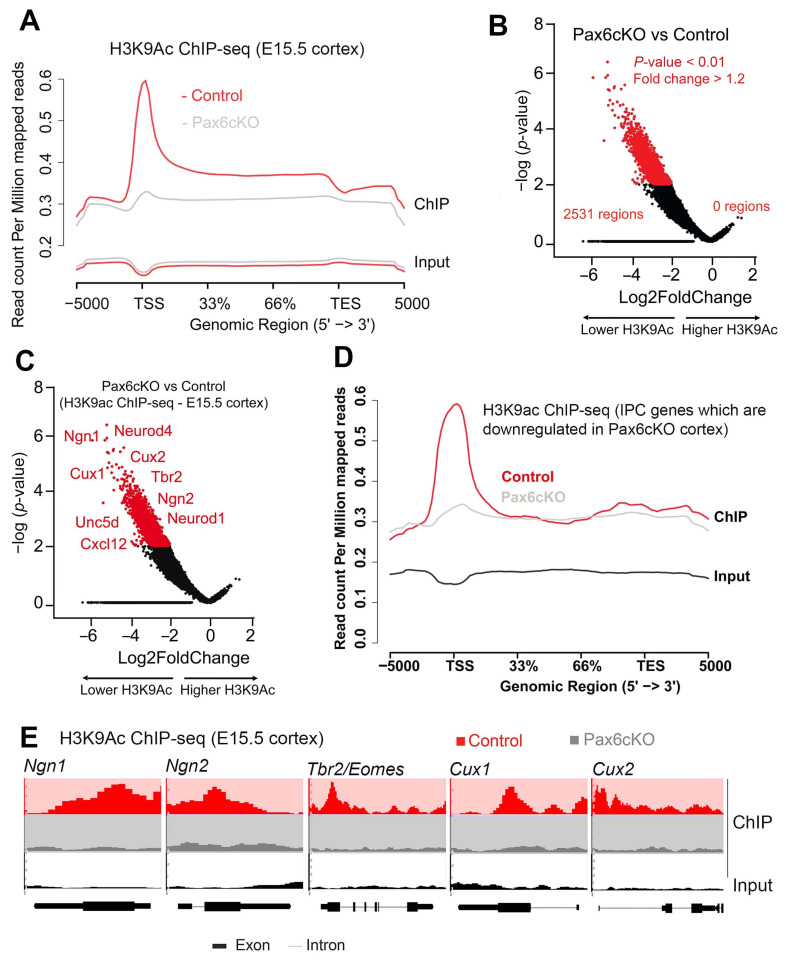
The H3K9ac level and neurogenic transcriptional program are disturbed in the absence of Pax6 during corticogenesis. (**A**–**D**) Analyzed ChIP-seq data showing H3K9ac distribution along pan-gene bodies in the E15.5 *Pax6cKO* cortex (**A**), significant changes (fold change (FC) > 1.2, paired Student’s *t*-test *p* < 0.01) in the association of H3K9ac with neurogenic genes in the E15.5 *Pax6cKO* cortex compared with wild-type (**B**,**C**), and the distribution of H3K9ac along IPC gene bodies in the E15.5 *Pax6cKO* cortex (**D**). (**E**) Distribution of H3K9ac along the IPC-enriched gene bodies of *Ngn1*, *Ngn2*, *Tbr2/Eomes*, *Cux1*, and *Cux2* in the *Pax6cKO* cortex (gray) versus control (red). Input (bottom row in black) and distributions after immunoprecipitation (top two rows) are indicated. Abbreviations: TSS, transcription start site; TES, transcription end site; IPCs, intermediated progenitor cells. Number of embryonic cortices used per group (*n*) = 3.

**Figure 5 biology-13-00068-f005:**
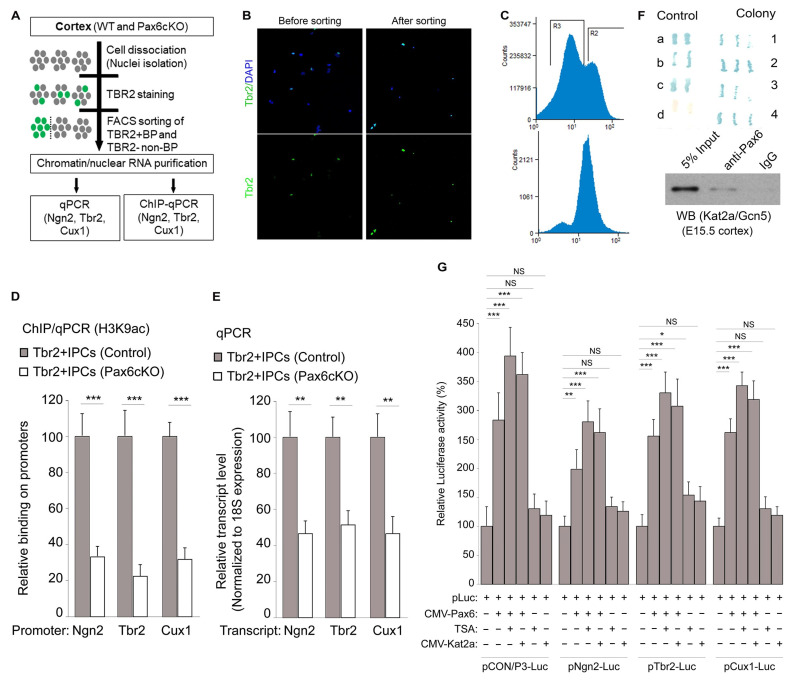
Pax6 expression is required for H3 acetylation-activated expression of Pax6 target genes in IPCs. (**A**) Schematic showing the experimental design used to sort TBR2-expressing cells in the *Pax6cKO* and wild-type cortexes for expression and promoter H3K9ac level comparison for selected bIP genes. (**B**) Micrographs showing cortical cells before and after isolation of TBR2-expressing cells. Counter staining with DAPI is shown. (**C**) Graphs showing the efficiency of TBR2+ cell sorting after the FACS experiment. (**D**,**E**) Bar graphs showing the promoter region level of H3K9ac (**D**) and the transcript levels (**E**) of the IPC-enriched genes, *Ngn1*, *Tbr2*, and Cux1, in the *Pax6cKO* cortex compared with the control. (**F**) Image showing the yeast two-hybrid assay, in which five different controls (a–d) were used in the performed two-hybrid yeast screen, as recommended by the manufacturer. Four colonies representing independent clones strongly positive for Pax6 binding revealed potential interactions between Gcn5/Kat2a and Pax6. (**G**) Luciferase assay demonstrating that Pax6 and H3 acetylation regulate the transcriptional program of Pax6 target genes. The promoter fragment DNA containing Pax6 binding site(s) of indicated Pax6 target genes were cloned into luciferase plasmids (Luc). Compared to control (Luc + CMV-EV), co-transfection of Pax6 (Luc + CMV-Pax6) was sufficient to activate luciferase activity. Presence of the H3 acetylation-augmenting conditions, TSA treatment, or transfection with CMV-Kat2A were important for the promoter activity of the indicated IPC genes (i.e., under *pNgn2/P3-Luc*, *pTbr2/P3-Luc*, and *pCux1/P3-Luc* conditions), leading to enhanced luciferase activity. Scale bar = 50 µm. Values are presented as means ± SEMs (* *p* < 0.05, ** *p* < 0.01, *** *p* < 0.005; NS, not significant; *n* = 6). Abbreviations: TSA, trichostatin A.

**Figure 6 biology-13-00068-f006:**
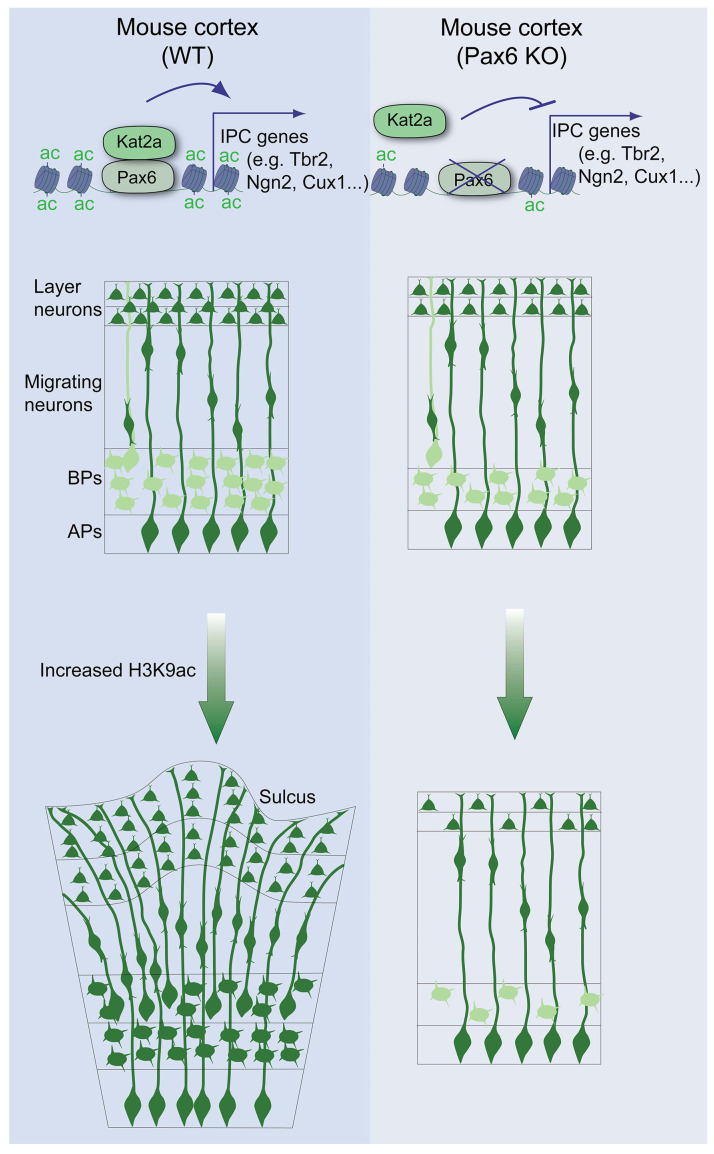
Schema summarizing the importance of Pax6 and H3 acetylation in the activation of IPC genes and attendant phenotypic effect in the developing cortex. In the wild-type cortex, Pax6 interacts with the H3 acetylation catalytic enzyme Kat2A to promote increased installation of H3K9ac at the promoter region of genes essential for IPC biogenesis. Thus, enhancing H3 acetylation accelerates the production of IPC in the developing cortex. This leads to expansion of the basal progenitor pool, which translates into an increase in neurogenesis, especially of upper-layer neurons, leading to cortical gyrification induction. However, in the absence of Pax6, H3K9ac levels drop, partly because of a lack of Kat2a interaction. This results in downregulation of IPC genes and a resultant reduction in upper-layer neuron production consequent to decreased IPC generation in the Pax6 mutant cortex.

## Data Availability

The original contributions presented in the study are included in the article/Appendix A, further inquiries can be directed to the corresponding author.

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
