# Peer review of "H3 Acetylation-Induced Basal Progenitor Generation and Neocortex Expansion Depends on the Transcription Factor Pax6"

_biology, 2024, doi:10.3390/biology13020068_

Round 1

Reviewer 1 Report (Previous Reviewer 2)

Comments and Suggestions for Authors

Dear Editor,

I have reviewed once more the paper by  Godwin Sokpor et al.

The authors have answered to my comments from the last revision.

I have only found that the figures in this version are blurry and need to be improved. 

Best regards

Author Response

Comment

I have only found that the figures in this version are blurry and need to be improved.

Response

We thank the Reviewer for the comment. The affected figures have been replaced with high quality versions. The improved figures are embedded in the revised manuscript.

Reviewer 2 Report (New Reviewer)

Comments and Suggestions for Authors

The study by Sokpor et al. build on previous work in the same lab to explore the interplay of Pax6 and histone acetylation in the development of the neocortex. Specifically, they find that Pax6 is required for the previously demonstrated increase of basal progenitor cells by H3 acetylation. Furthermore, their results indicate that Pax6 interacts with the HAT Kat2a, which specifically mediates H3K9 acetylation.

This is a well-conducted study. Experiments are clearly laid out and well documented. The results provide the evidence that supports the findings.

I only have a number of minor points that should be addressed:

·        Lines 66/67: What does ‘IPC’ refer to? The abbreviation is not explained. Do you mean all intermediate progenitor cells, or specifically bIPC?

·        Lines 74-76: “when Pax6 expression is protracted in the daughter cells of Tis21-expressing aRGCs, it promoted cell cycle re-entry with attendant increase in the proliferation of such BP derivatives” – this is somewhat misleading since it either implies that aRGCs are derived from BPs or that aRGCs are BPs.

·        Lines 84-96: The Narayanan (2018) paper suggested that the BAF155 knockout results in a shift from aRGCs to bRGCs and a reduction of bIPC. This could be phrased more clearly in the paragraph.

·        Line 243: The abbreviation ‘IZ’ should be explained at first use. It would also be good to mention the relationship of the IZ with VZ, SVZ and the different progenitor populations in the introduction.

·        Fig. 2A: The annotation lists WT+HDACi and BAF155cKO+TSA. This is a little confusing, since the HDAC inhibitor used on the wildtypes was also TSA. Better to be consistent in the labelling.

·        Fig. 2E: The annotation states ‘Number of RG cells in…’ – should this be bRG cells (or does this include aRGC/NSC?)?

·        Fig. 3D: It would be good to maintain a consistent order/colour/shading of columns in all figures. Fig. 3D either has a different order (it should be control, control + TSA, BAF155cKO, BAF155cKO + TSA, BAF155_Pax6_dcKO, BAF155_Pax6_dcKO + TSA like in Fig. 3C) or the shading of control + TSA and BAF155cKO is mixed up.

·        Fig. 4: The C and D labelling should be the other way around.

·        Lines 394ff.: What was the rationale to specifically investigate H3K9Ac, rather than other chromatin marks like H3K27Ac?

·        Line 409: What do you mean by ‘expression’ in this context? You have assayed for the presence of a post-translational modification, not expression.

·        Line 474-477 (and Fig. 5G): This could be phrased more clearly. Fig 5G indicates that any increase of luciferase activity beyond background depends on Pax6. Kat2A or TSA treatment alone are not sufficient to activate reporter expression (except for a slight increase in the pTbr2-Luc construct). Also, was the increase in luciferase activity in presence of Kat2A or TSA in addition to Pax6 statistically significant? Is the assumption that the transfected constructs are integrated or that the plasmid DNA recruits histones? The alternative could be that the increased acetylation activates the expression of other transcription factors to indirectly increase luciferase activity.

·        Lines 481-489: The second part of this paragraph and Fig. 6 would be better suited for the discussion section.

·        Lines 526-529: The evolutionary argumentation is not quite clear. Are you implying that there is less H3 acetylation in non-primate mammals? Is there direct evidence for this?

·        Methods: This section seems to be missing the transfection and luciferase assay used for Fig. 5G, which should be added.

Specific text corrections:

·        Line 42: …, a subpopulation of APs giveing

·        Line 53: …, proliferative bRGCs make up a

·        Line 76: the sustained expression of Pax6- induced generation (remove hyphen)

·        Line 97: acetylation is an importantce epigenetic

·        Line 296: We observed even higher levels of expression of these bRGC

·        Lines 518/519: Our recent studyies (Kerimoglu et al. 2021), including thisand the current one, adds a critical layer

Author Response

We thank the Reviewer for the extensive and constructive comments and suggestions. All concerns have been addressed leading to an improvement in the quality of the manuscript.

Comment

Lines 66/67: What does ‘IPC’ refer to? The abbreviation is not explained. Do you mean all intermediate progenitor cells, or specifically bIPC?

Response

We have defined IPCs to stand for intermediate progenitor cells (Line 74)

Comment

Lines 74-76: “when Pax6 expression is protracted in the daughter cells of Tis21-expressing aRGCs, it promoted cell cycle re-entry with attendant increase in the proliferation of such BP derivatives” – this is somewhat misleading since it either implies that aRGCs are derived from BPs or that aRGCs are BPs.

Response

We thank the Reviewer for pointing this out. We have amended the sentence (Lines 81-83) to make it clear.

Comment

Lines 84-96: The Narayanan (2018) paper suggested that the BAF155 knockout results in a shift from aRGCs to bRGCs and a reduction of bIPC. This could be phrased more clearly in the paragraph.

Response

The sentence (Lines 95-96) has been changed to focus on the bRGC effect occurring at the expense of neuronal differentiation.

Comment

Line 243: The abbreviation ‘IZ’ should be explained at first use. It would also be good to mention the relationship of the IZ with VZ, SVZ and the different progenitor populations in the introduction.

Response

We have now defined IZ at the part of the text it’s first used (Line 241).  We have briefly described the cortical regions and other concepts to keep the introduction concise. The references provided give more context and description. For instance (Lui et al. 2011; Borrell and Gotz 2014; Taverna et al. 2014; Dehay et al. 2015; Llinares-Benadero and Borrell 2019; Molnar et al. 2019) offer excellent information on the cortical structure and regions.

Comment

Fig. 2A: The annotation lists WT+HDACi and BAF155cKO+TSA. This is a little confusing, since the HDAC inhibitor used on the wildtypes was also TSA. Better to be consistent in the labelling.

Response

The correction has been effected.

Comment

Fig. 2E: The annotation states ‘Number of RG cells in…’ – should this be bRG cells (or does this include aRGC/NSC?)?

Response

The correction has been done.

Comment

Fig. 3D: It would be good to maintain a consistent order/colour/shading of columns in all figures. Fig. 3D either has a different order (it should be control, control + TSA, BAF155cKO, BAF155cKO + TSA, BAF155_Pax6_dcKO, BAF155_Pax6_dcKO + TSA like in Fig. 3C) or the shading of control + TSA and BAF155cKO is mixed up.

Response

The keys are the same for both figures. This is so presented in C because we want to show the basal difference between BAF155cKO and Wildtype used as controls following Veh treatment. Thus, putting them side by side makes them easily comparable graphically.

Comment

Fig. 4: The C and D labelling should be the other way around.

Response

We thank the Reviewer for pointing this out. The order of the labelling has been corrected.

Comment

Lines 394ff.: What was the rationale to specifically investigate H3K9Ac, rather than other chromatin marks like H3K27Ac?

Response

It is because we found in our previous work (Kerimoglu et al. 2021) that H3K9Ac upregulation induces BP genesis and it is enriched in BP found in primate cortex. The sentence has been modified to remind readers of such backdrop.

Comment

Line 409: What do you mean by ‘expression’ in this context? You have assayed for the presence of a post-translational modification, not expression.

Response

Indeed, it’s a post-translational modification of gene distribution analysis done. We have modified the sentence by removing the term ‘‘expression’’.

Comment

Line 474-477 (and Fig. 5G): This could be phrased more clearly. Fig 5G indicates that any increase of luciferase activity beyond background depends on Pax6. Kat2A or TSA treatment alone are not sufficient to activate reporter expression (except for a slight increase in the pTbr2-Luc construct). Also, was the increase in luciferase activity in presence of Kat2A or TSA in addition to Pax6 statistically significant? Is the assumption that the transfected constructs are integrated or that the plasmid DNA recruits histones? The alternative could be that the increased acetylation activates the expression of other transcription factors to indirectly increase luciferase activity.

Response

As shown in the results (Figure 5G), both conditions of Pax6+Kat2A and PaX6+TSA significantly (***) increased the luciferase activity. We found that luciferase activity remained at baseline in the presence of only Kat2A. This observation sets the background for arguing that no observable indirect effect influenced the luciferase activity.

Comment

Lines 481-489: The second part of this paragraph and Fig. 6 would be better suited for the discussion section.

Response

We have removed the paragraph as similar information is given in the discussion.

Comment

Lines 526-529: The evolutionary argumentation is not quite clear. Are you implying that there is less H3 acetylation in non-primate mammals? Is there direct evidence for this?

Response

Due to the absence of direct evidence, the sentence (Lines 507-510) has been rephrased as a question asking for such possibility.

Comment

Methods: This section seems to be missing the transfection and luciferase assay used for Fig. 5G, which should be added.

Response

The luciferase assay has now been added to the methods.

Specific text corrections:

Line 42:

Response

The correction has been done (Line 55)

Line 53:

Response

The correction has been done (Line 64)

Line 76:

Response

The correction has been done (Line 83)

Line 97:

Response

The correction has been done (Line 102)

Line 296:

Response

The correction has been done (Line 291)

Lines 518/519:

Response

The correction has been done (Lines 500-501)

Reviewer 3 Report (New Reviewer)

Comments and Suggestions for Authors

The article by Sokpor et al. is a follow-up from their 2021 study which showed that histone H3 lysine 9 acetylation (H3K9ac) is low in murine bIPs and high in human bIPs. Suggesting that H3K9ac increases bIP proliferation, increasing the size and folding of the normally smooth mouse neocortex. H3K9ac drives bIP amplification by increasing the expression of the evolutionarily regulated gene, Trnp1, in the developing cortex. They described an unknown mechanism that controls cortical architecture.

Now, in this new study, they are using a large number of techniques to evaluate the contribution of H3 acetylation in the Pax6 function in basal progenitor genesis and cortical expansion. Their new data shows that the event of basal progenitor generation and cortical expansion is enhanced by the increased H3 acetylation in the Pax 6 presence.

General comments

The quality of the writing is generally good throughout, the text is clear, and the paper will be of interest to readers of Biology. The gap in the literature that motivates the study is clearly presented in the introduction, the results are clearly presented, and the conclusions appear sound.

Main and major comment:

Several images and grafts in Figures 1 and 3 are taken from their previous study by Sokpor et al. (2021).

Publications must include new data and new images. Authors must replace the images used in their previous study with new ones.

Minor comments:

To facilitate the compression of the results included in Figure 2, authors should reorganize this figure and maintain it in order A, B, D,…..Bar graphs (E) should go at the end of the Figure.

Author Response

We thank the Reviewer for the constructive comments and suggestions. Most of their concerns have been addressed leading to an improvement in the quality of the manuscript.

Main and major comment:

Several images and grafts in Figures 1 and 3 are taken from their previous study by Sokpor et al. (2021). Publications must include new data and new images. Authors must replace the images used in their previous study with new ones.

Response

As part of our publication agreement, permission has been given for reuse of parts of our previously publish work (Kerimoglu et al., 2021), which forms the basis of the current work. We refer to the permission under a Creative Commons Attribution NonCommercial License 4.0 (CC BY-NC). We have cited the previous work in the corresponding figure legends.

Minor comments:

To facilitate the compression of the results included in Figure 2, authors should reorganize this figure and maintain it in order A, B, D,…..Bar graphs (E) should go at the end of the Figure.

Response

The reviewer’s suggestion is ideal, however, the nature of the figures give less flexibility to make the rearrangement. We have tried to maintain the alphabetical order as much as possible while preserving the logic of have the quantification in A following the relevant micrograph in B and the micrographs in C/D preceding their quantifications in E.

Round 2

Reviewer 3 Report (New Reviewer)

Comments and Suggestions for Authors

Several images and grafts in Figures 1 and 3 are taken from their previous study by (Kerimoglu et al., 2021). Publications must include new data and new images. Authors must replace the images used in their previous study with new ones

Author Response

Comment

Several images and grafts in Figures 1 and 3 are taken from their previous study by (Kerimoglu et al., 2021). Publications must include new data and new images. Authors must replace the images used in their previous study with new ones

Response

We thank the Reviewer for the comment. The approach was adopted because of difficulty retrieving images from our raw data file due to lab relocation. Fortunately, the co-author responsible for data curation has made available new images with which the figures have been modified. The new images are from other brain sections obtained from the same animals used in the previous study. The revised figures have been included in this round of revision as Figure 1_revised and Figure 3_revised. The current version of the manuscript has been amended accordingly.

This manuscript is a resubmission of an earlier submission. The following is a list of the peer review reports and author responses from that submission.

Round 1

Reviewer 1 Report

Comments and Suggestions for Authors

Main concerns

1) I have an issue with the way that the authors are portraying the importance of H3 acetylation for neural progenitors. In the Introduction, they state (line 216): "We recently found that increased H3 acetylation specifically promotes genesis of BPs, including both bIPCs and bRGCs (Kerimoglu et al. 2021)."

The implication (not explicitly stated) is that H3ac increases lead to the production of basal progenitors instructively. However, 1) there isn't any evidence that H3ac levels are dynamic in apical progenitors (ie. prior to BP production). 2) The manipulation is actually a drug that blocks histone deacetylases (which do many things beyond regulating histone acetylation - eg. they deacetylate other proteins, some have cytoplasmic functions etc.). This also means that H3ac is not added to the genome by the authors' manipulations. Instead, it fails to be taken away, implying that imbalancing the turnover of acetylation on genes leads to the reported effects.  I think that the paper should frame these issues more precisely for the readers in the abstract/introduction, and throughout the paper. Eg. for figure 1, rather than "acetylation induces BPs", it should be "deacetylases impair BPs".

2) M&M section is incomplete. Animal work was not described. Ethics protocols not presented. Injections, in utero electroporations etc. not described.

3) Student's t-test is not valid for the comparisons presented in many of the graphs. These tests should be ANOVA + Tukey's or ANOVA + Dunnett's. Also, the n-values need to be reported somewhere - preferably by displaying them directly in the graphs.

4) Figure 1 shares many identical panels/data with Figure 3 of Kerimoglu et al.  I have the impression that the same is true for Figure 3 and Figure 7 of Kerimoglu et al. I presume that the authors meant to cite the previous paper. This needs to explained explicitly for the reader - otherwise it might lead to retraction.

Reviewer 2 Report

Comments and Suggestions for Authors

I have reviewed the manuscript by Godwin Sokpor et al. I have found that their results are conclusive.

These authors elegantly demonstrate by using several biological tools (immunohistochemistry, RNA sequencing, Chromatin immunoprecipitation and sequencing, and yeast two-hybrid assay) the mechanistic basis of basal progenitor (BP) pool expansion. The authors show that the presence of Pax6 is essential for direct interaction with Kat2a to cause H3K9ac-linked activation of BP genes.

I do have some recommendations to the authors that I believe will improve the manuscript.

11- In the result section, I have found excessive discussion of each experiment. I would limit the result section just by introducing the purpose of the experiment and leave the discussion of the topic to the discussion section or include this information in the introduction so as to prepare the readers for the result section.

22-  There is no information in the introduction to why the authors use the mutant BAF155KO in their experiments. Please add a sentence or two to refer to the selection of this mutant.

33-  Many abbreviations are used with no reference to their meaning, for example NSC is presented in the text with no indication to it meaning “neuronal stem cells”. The same applies to BAF.

44-  The font/size of the text is different in lines 35 (depletes the BP pool) and 174 ((http://bioinf.wehi.edu.au/featureCounts/) .

55-  Please cite Image J software.

66-  Line 92.  a stop mark is missing before “This”.